# Accelerating process development for 3D printing of new metal alloys

David Guirguis [1,2] ✉, Conrad Tucker [1,2,3] & Jack Beuth[1,2]

Addressing the uncertainty and variability in the quality of 3D printed metals can further the wide spread use of this technology. Process mapping for new alloys is crucial for determining optimal process parameters that consistently produce acceptable printing quality. Process mapping is typically performed by conventional methods and is used for the design of experiments and ex situ characterization of printed parts. On the other hand, in situ approaches are limited because their observable features are limited and they require complex high-cost setups to obtain temperature measurements to boost accuracy. Our method relaxes these limitations by incorporating the temporal features of molten metal dynamics during laser-metal interactions using video vision transformers and high-speed imaging. Our approach can be used in existing commercial machines and can provide in situ process maps for efficient defect and variability quantification. The generalizability of the approach is demonstrated by performing cross-dataset evaluations on alloys with different compositions and intrinsic thermofluid properties.

Additive manufacturing (AM) can be considered one of the pillars of the fourth industrial revolution. The industry has the potential to play a major role in innovation processes and in the US and global economy[1]. Metal AM is becoming essential in many industries, including healthcare, aerospace, and defense, due to the benefits of lead time reduction, enhanced production efficiency, part consolidation, and design freedom. Laser powder bed fusion (L-PBF) is the most widely used technology for printing metal alloys. The technology uses a high-power laser as an energy source to melt and fuse powders in specific locations to form certain shapes, a recoater then spreads a new layer of powder, and the process repeats until 3D objects are formed.

The variability problem is the main obstacle that hinders the reliability of the quality of printed parts and thus the potential for full production. The mechanical properties and dimensional accuracy of printed parts vary depending on the powder and machine used, the scanning strategy, and the printing conditions[2–4]. The lack of repeatability and uncertainty in quality has motivated many researchers to better understand the process, the influence of decisive parameters, and process-property relationships to find ways to control the quality and microstructure properties of printed objects[2,5–8]. Moreover,

mapping process parameters into printing defects is essential to determine the optimal process parameters for each type of metal alloy and printing facility. Process development is typically performed by trained laboratory technicians using ex situ facilities where printed tracks are characterized[9]. The laser beam power and velocity are two major machine parameters that directly control the laser energy density and, therefore, the stability of the molten pool of metal[10,11]. Additionally, adequate spacing between laser scan tracks should be determined to address the variability and uncertainty in the width of the printed tracks so that sufficient overlap between each pair of melted and fused tracks can be achieved to prevent residual unmelted powder. This type of defect can deteriorate the mechanical properties and reduce fatigue life[12,13].

High-speed imaging by a laser at a fine spatiotemporal scale has been used to monitor molten pools. Although this monitoring technique has been used in previous work, adopting it for in situ process mapping is limited due to the poorly observable features[14], low algorithmic accuracy[15], and the need to use a complex setup and installation to obtain temperature field measurements[16,17], which may require imaging alignment, calibration, and a special setup that needs to be integrated with the scanning head of the 3D printing machine.

[1]Next Manufacturing Center, Carnegie Mellon University, Pittsburgh, PA, USA. [2]Mechanical Engineering Department, Carnegie Mellon University, Pittsburgh, PA, USA. [3]Machine Learning Department, Carnegie Mellon University, Pittsburgh, PA, USA. ✉e-mail: dguirguis@cmu.edu

In this work, we develop an in situ approach for designing an accelerated and efficient process for 3D printing of new metal alloys to achieve melt-pool stability and address the variability problem. Based on the advancement of knowledge in defect formation dynamics[11,18–20], we devise an approach to eliminate saturation in frames captured by a high-speed camera to monitor the dynamic changes in a molten pool and use temporal data to classify the process into different types of defects by using state-of-the-art video vision transformers[21] rather than convolutional neural networks (CNNs) and traditional computer vision approaches for static images[15,22]. This approach can enhance the algorithmic accuracy to over 98% without the need to use an advanced setup with high-cost pyrometers to extract the temperature field[16,23]. Another important aspect for the deployment of in situ systems is their generalizability to new alloys. Thus, we test our approach on metal alloys that are not used to train the vision transformer model and achieve a top-1 accuracy of up to 98%. In addition, to address the variability problem, we generate variability process maps for molten pool attributes to guide the determination of optimal hatch spacing. The pipeline of our method is depicted in Fig. 1.

## Results

### Capturing melt-pool dynamics

We developed an off-axial imaging setup at the L-PBF facility of the CMU Next Manufacturing Center, which consists of a high-speed camera and magnification lenses with optical filters attached to it. The optical train is devised to block the wavelength that is associated with most of the emissions of the plasma plume, i.e., ionized vapor, condensed particles, and plumes formed during printing. The plume temperature alone can be higher than 3500 K[17,24]. The representative results of melt-pool frames captured for different printing regimes are shown in Fig. 2. The videos are recorded at a high rate of 54,000 frames per second to capture the high-frequency oscillation in the melt-pool

shape. The gradients of the melt-pool light emission are clearer than those in frames captured by other direct imaging setups[15,25,26]. In addition, the melt-pool geometric attributes are intuitively reasonable and are matched with frames captured with setups calibrated for temperature measurements and imaging setups with illuminated scenes; see e.g., [27].

The melt pool becomes smaller but elongates as the scan speed increases. However, melt pools captured in the lack-of-fusion regime are very small with a low length-to-width ratio, as the energy density is very low and the laser beam does not penetrate deeply into the material. Although the captured melt pools are clearly distinct from the plasma and plume, they are still observable with high oscillations in the keyholing regime. In the keyholing regime, owing to the high energy density, the vapor plasma has a significantly high energy density and cannot be easily filtered out. Examples of videos for the four regimes are presented in Supplementary Movies S1–S4. In the balling regime, in agreement with the modeling study[28,29], the molten pool elongates and disconnects, leaving behind peaks in the track. A post-processed frame extracted from a video of a P-V combination that is known to be associated with the balling regime is shown in Fig. 2c.

To further investigate the dynamic changes in melt-pool shapes that can be captured by the imaging setup, we analyzed the changes in the maximum intensity of the captured melt-pool emissions and the cross-correlation of subsequent frames. As shown in Fig. 3a–c, melt pools with severe keyholing have the highest fluctuation range, whereas tracks printed in conduction modes have more stable maximum intensity values. The keyhole fluctuation is typically characterized by fluctuations in the width and depth of the keyhole[11,30–32]. The keyhole depth fluctuation is reported in previous studies as up to approximately 10 kHz[32]. However, the intensity fluctuation is found to have a higher frequency of approximately 8–17 kHz. Notably, the intensity fluctuation does not directly represent the fluctuation in the

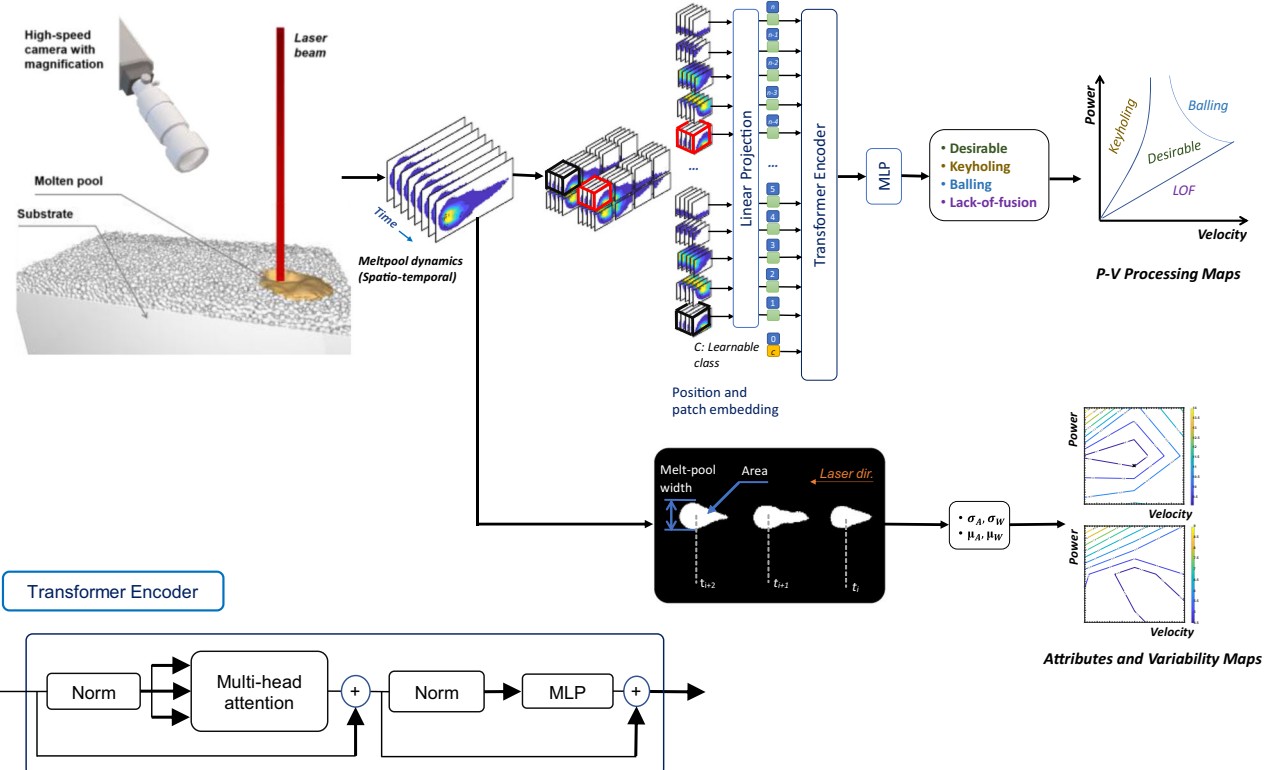

**Fig. 1 | Pipeline of the proposed in situ process development approach.** A high-speed imaging setup is used to monitor the dynamic changes in the molten pool, and the spatio-temporal data is used to classify the process into different types of defects and printing regimes using video vision transformers. The variability in the morphological attributes of the molten pool is captured from the imaging data and processing maps of variability, represented by the standard deviations, are constructed indicating the processing parameters that can result in a more stable process.

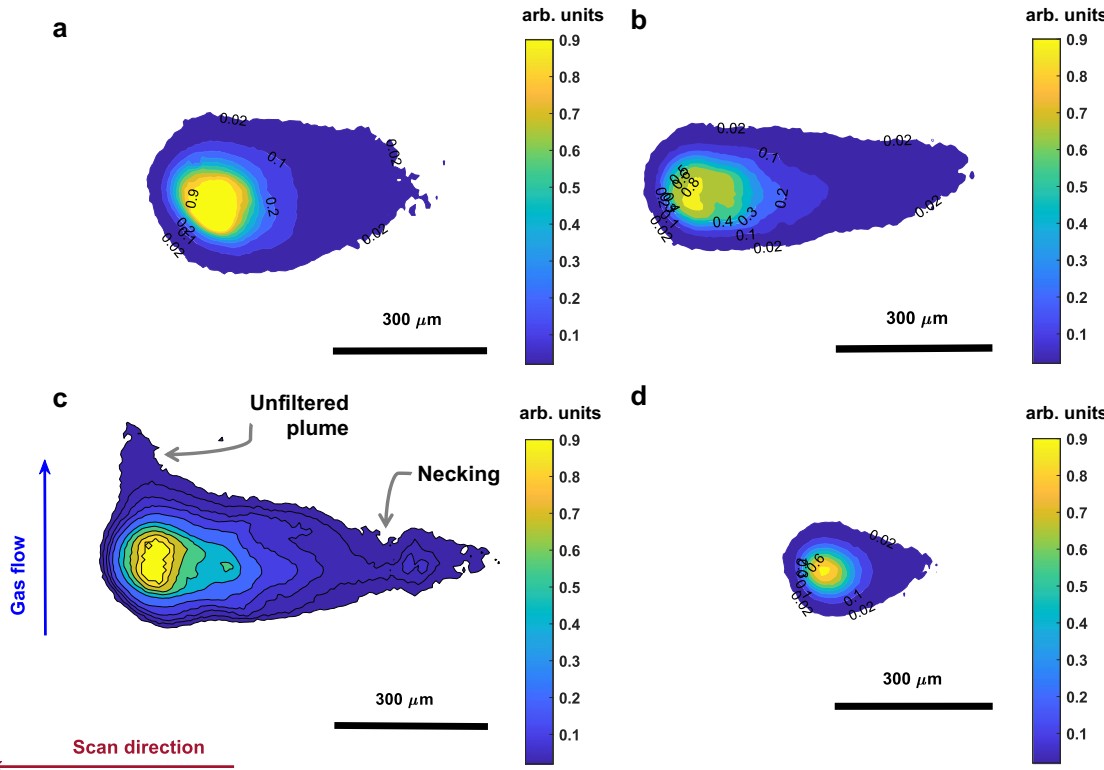

**Fig. 2 | Imaging of molten pools captured at different printing regimes.** Images captured at 54,000 frames per second during printing of Ti-6Al-4V tracks in different processing regimes. **a** Keyholing (350 W, and 600 mm/s), (**b**) Stable (320 W, and 1200 mm/s), (**c**) Balling (400 W, and 1800 mm/s), (**d**) Lack-of-fusion (150 W, and 1500 mm/s).

keyhole depth temperature because the emissions of the plasma plume can block the view of the keyhole depth.

The temporal changes in the melt-pool shapes are qualitatively analyzed by calculating the correlation coefficient between subsequent frames. Box plots of the correlation coefficients calculated for melted beads at different regimes are illustrated in Fig. 3d–f. To depict how the melt-pool changes over time at different energy densities, the average values are plotted in the power-velocity (P-V) map, as shown in Fig. 3g–i.

## Vision transformer model development

Transformers are state-of-the-art self-attention deep learning architectures for natural language processing. Vaswani et al. [33] demonstrated that pure transformers without recurrent or convolutional layers can overcome challenges, such as vanishing gradients in long-range sequences and the inability to perform operation in parallel, that other sequence modeling approaches face. Recently, ref. [34] demonstrated the effectiveness of pure transformers in computer vision and achieved outstanding results compared to the results achieved in other studies on CNNs. ViTs have less image-specific inductive bias than CNNs[34]. In addition, methods that embed input and slice videos into nonoverlapping patches without embedding their positions can make the input more suitable for the problem at hand.

The transformer layers consist of layer normalization, multiheaded self-attention, a multilayer perceptron (MLP) of linear projects, and Gaussian error linear unit (GELU[35])[33,34]. The captured videos are postprocessed and divided into nonoverlapping 3D patches across the temporal and spatial domains and then linearly projected along with the positional embedding of the patches to the transformer encoder, as depicted in Fig. 1. This method of extracting and feeding the temporal-spatial information to the model is called tubelet embedding and was proposed by Arnab et al. [21].

Due to the small field of view of the high-speed camera and the high scan speed of the laser, a limited number of frames are stored in each recorded video. Thus, we implement regularization schemes to efficiently deal with small amounts of data. Biases and layer weight regularization are applied to the MLP of the multiheaded transformers and the MLP head. Although data augmentation is powerful in boosting the performance of transformers[36], we rely instead on regularization to preserve dynamic changes in the molten pool that are reflected in the image intensity and changes in the geometrical attributes.

## Process parameter mapping

The videos are classified into four categories: desirable regimes and printing regimes with three different types of defects: keyholing, balling, and lack-of-fusion. Keyholing is deep drilling into the material through vaporization, and it results in a deep vapor cavity[37]. Although keyholing can occur in other printing regimes, in this context, we refer to keyholing defects, which are characterized by unstable, deep, and narrow penetration and can lead to enclosed pores inside the printed parts. These cavities can result in cracks and thus can degrade the fatigue life of the parts[38]. Another type of defect is balling, which is also known as humping in welding literature[39]. In the balling regime, owing to Plateau–Rayleigh instability and Marangoni flow, the melted tracks exhibit a rough surface with a periodic ball cross-section shape and are associated with undercuts at the corners. The last class of defects is lack-of-fusion, where the energy density is not sufficient to fully melt the powder, so unmelted powder and irregular gaps are observed between the melted tracks. Examples of the four printing classes are illustrated in Fig. 4.

We conducted single-bead experiments with different P-V combinations, covering the four printing regimes, on stainless steel SS316L, titanium alloy Ti-6AL-4V, and Inconel alloy IN718. To explore the generalizability of the method, we performed a cross-dataset

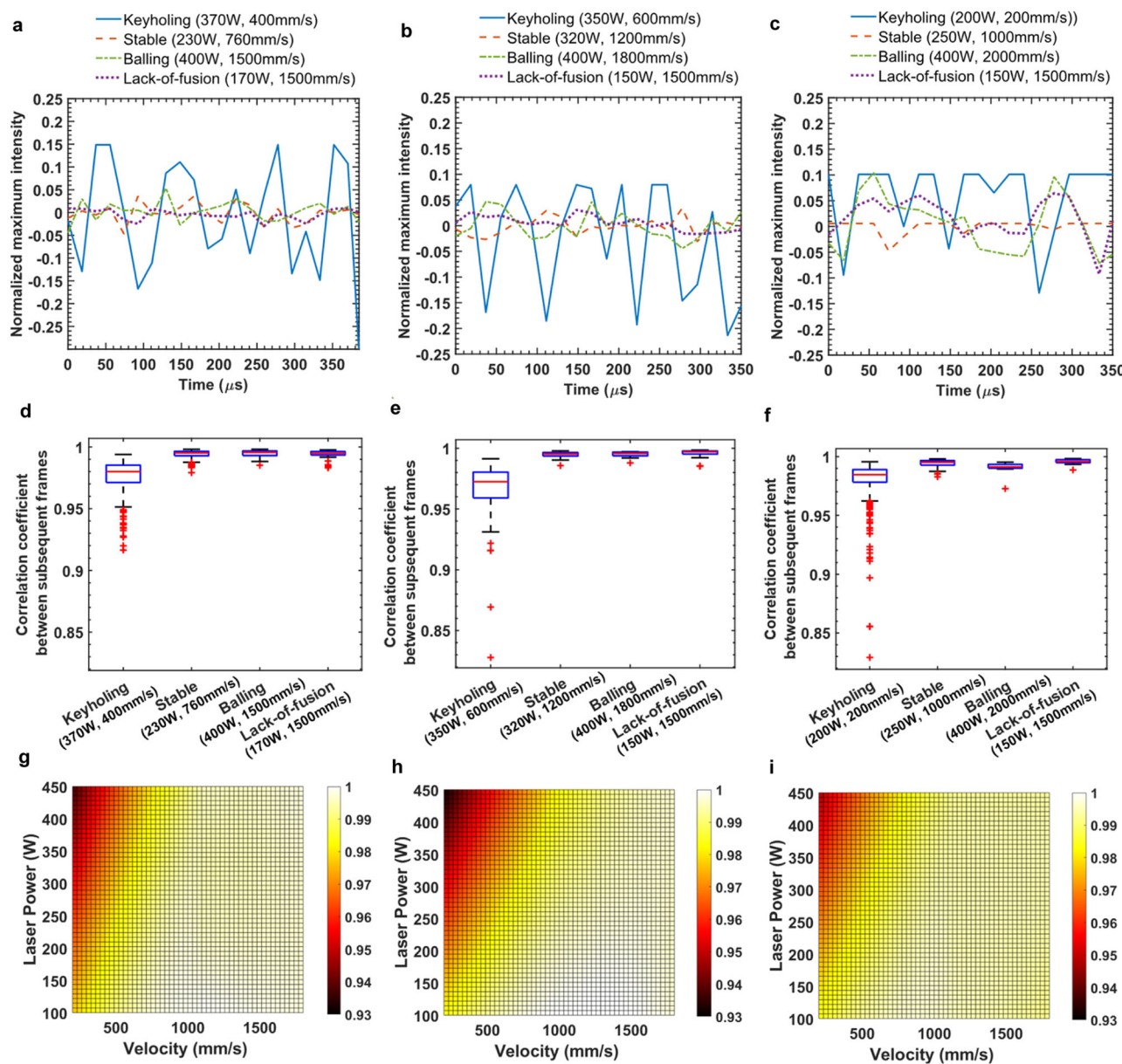

**Fig. 3 | Quantification of molten pool surface dynamics.** Left to right: IN718, Ti-6Al-4V, and SS316L. **a–c** Normalized maximum intensity of melted tracks for different regimes. The maximum intensity values of the images are normalized with zero mean and one standard deviation. **d–f** Correlation coefficients between subsequent frames of recorded videos. **g–i** P-V map of melt-pool surface changes depicted by the average mean values of the cross-correlation coefficients between subsequent frames. The surface plots are generated by using local linear regression. Source data are provided as a Source Data file.

evaluation, where the model is trained on the recorded videos of one alloy and tested on the videos others while the hyperparameters are kept unchanged. The classification results of the training experiment on SS316L alloy are listed in Tables 1 and 2. The top-1 and top-2 accuracies obtained by running inference on the IN718 alloy dataset are 96.63% and 100%, respectively, whereas the achieved accuracies on

the Ti-6AL-4V alloy data are 87.60% and 95.87%, respectively. The F-1 score, which is a combined measure of recall and precision of classification, ranges from 0.69 in Ti-6Al-4V balling to 1.0 in the case of IN718.

The classification accuracy for the Ti-6Al-4V data is expected to be lower than that of the IN718 alloy data. The Ti-6Al-4V alloy is observed to emit a denser vapor plume in comparison to the other alloys. The alloy elements of Ti-6Al-4V experience significant vaporization in comparison to those of SS316L and IN718[40]. Moreover, Ti-6Al-4V has lower thermal conductivity, higher absorptivity to laser radiation and much different thermophysical properties from the other alloys[41].

After classification, the classes of the P-V combinations are averaged across the samples in the testing datasets and plotted to generate process maps. The process maps[42–44] or printability maps are maps of the resultant printing outcome for the P-V combinations. Power and velocity are the two main controllable processing parameters that

**Table 1 | Results of defect and processing regime prediction by training on SS316L and testing on IN718**

|                | Precision | Recall | F1-score | Samples |
|----------------|-----------|--------|----------|---------|
| Desirable      | 0.99      | 1.00   | 0.99     | 428     |
| Keyholing      | 1.00      | 0.98   | 0.99     | 334     |
| Balling        | 1.00      | 1.00   | 1.00     | 213     |
| Lack-of-fusion | 0.99      | 0.99   | 0.99     | 568     |

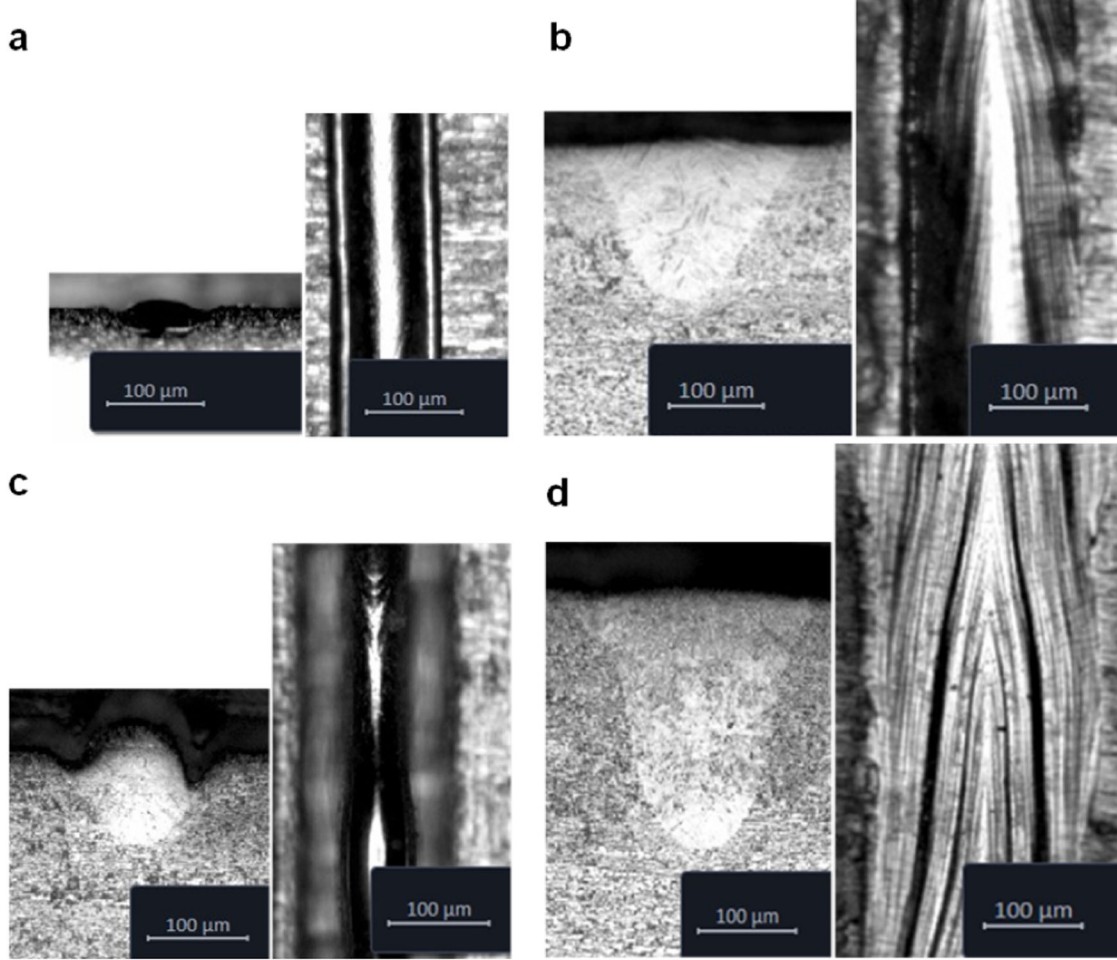

**Fig. 4 | Examples of the 3D printing regimes.** Micrographs of sectioned beads and top views of printed single beads that represent the four printing classes. **a** lack-of-fusion (150 W, 1500 mm/s), (**b**) desirable (350 W, 1200 mm/s), (**c**) balling (400 W, 1800mm/s), and (**d**) keyholing (350 W, 600 mm/s). The samples are printed in Ti-6Al-4V.

determine the input energy density and therefore have a major influence on the dynamics of the molten pool and its stability[8]. Figure 5 shows the process maps generated by our in situ method with the vision transformer model. The process maps generated by our method are found to be in good agreement with the maps generated after ex situ characterization of printed beads (see Materials and Methods), except for a misclassified point with high laser power in the balling regime of Ti-6Al-4V.

Note, however, that there is no clear separation between these classes[45]. Keyholing may occur in desirable tracks and coexist with balling without forming pores. Moreover, balling can occur in shallow melt pools[29] as well as in melt pools with elongated keyholes[46]. For instance, as shown in Fig. 5b, although the printed beads at 220 W and 1400 mm/s are very shallow and can lead to lack-of-fusion defects, they are labeled in the current study as balling since beading is observed on

the track surface. The detailed training results on IN718 and Ti-6Al-4V alloys are included in Supplementary Tables S1–S4. The validation results and confusion matrices are shown in Fig. 6. To test the influence of random initiation, the experiments are repeated five times with a different random seed and shuffled training data. Bar plots of the classification accuracies are illustrated in Fig. 7. The experiments show a significant influence of the initiation as the model is trained from scratch and the training data are shuffled in each experiment. However, the accuracy values are still reasonably high.

The effect of the backbone capacity on the performance is illustrated in Fig. 8. Although the accuracy can be improved by increasing the backbone capacity, sufficient temporal context is necessary to extract meaningful patterns for the dynamics of the melt pool. In this particular classification problem, the model distinguishes the same object and the same action but with different topological shapes and dynamics. On the other hand, in most other classification problems, the task is to classify different objects or distinct actions.

To validate the performance of the method, experiments are performed to compare the video vision transformer with other state-of-the-art models. Two pretrained video vision transformer models, ViViT-B[21] and TimeSformer[47], are compared with a deep convolutional network (VGG16)[48], a deep residual model (ResNet152)[49], and a mobile video network model (MoViNet-A1)[50]. As the results in Table 3 show, the video vision transformer models outperform the CNN-based

**Table 2 | Results of defect and processing regime prediction by training on SS316L and testing on Ti-6AL-4V**

|               | Precision | Recall | F1-score | Samples |
|---------------|-----------|--------|----------|---------|
| Desirable     | 0.68      | 0.86   | 0.76     | 166     |
| Keyholing     | 0.94      | 0.98   | 0.96     | 441     |
| Balling       | 1.00      | 0.52   | 0.69     | 142     |
| Lack-of-fusion | 0.97     | 1.00   | 0.98     | 291     |

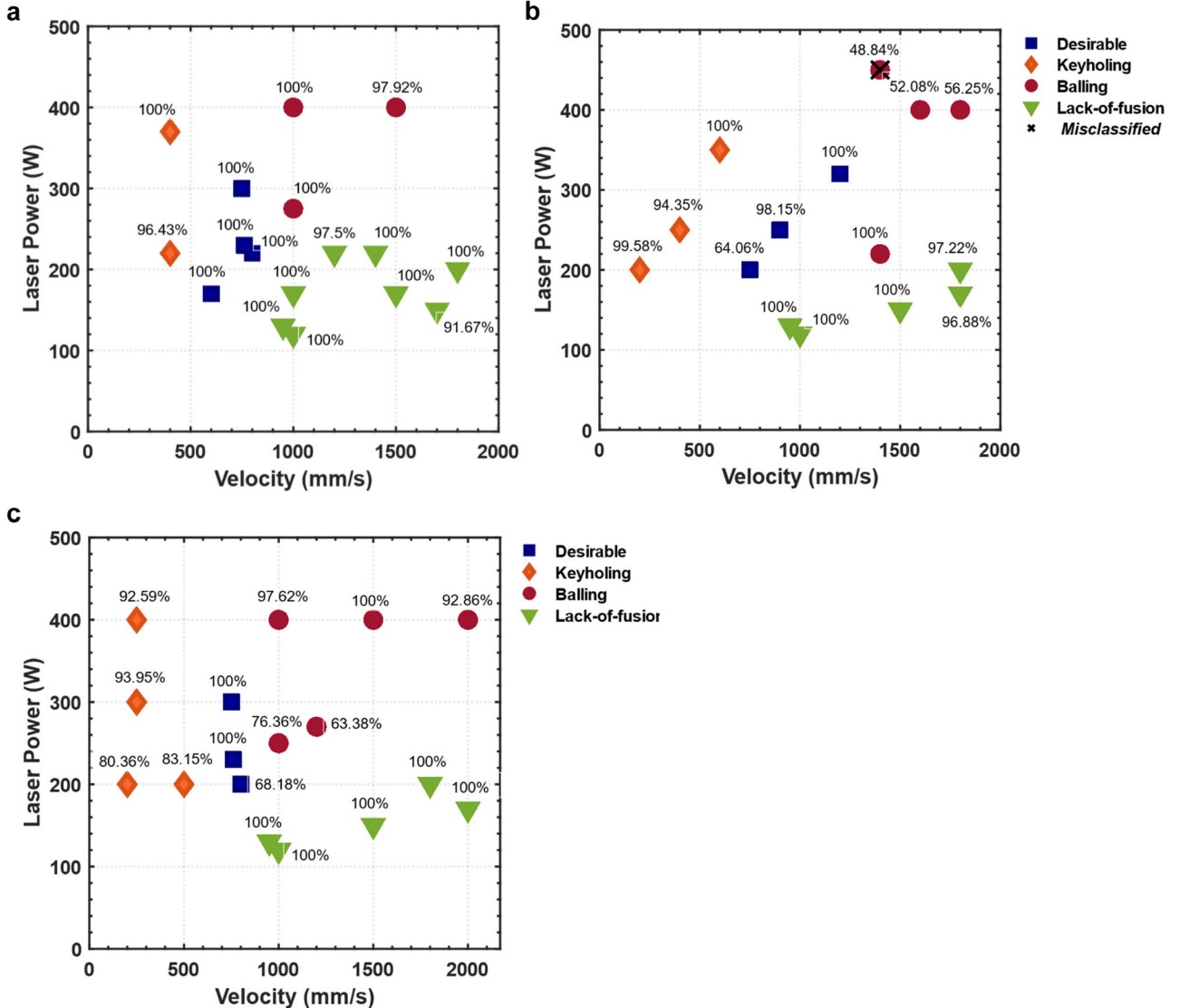

**Fig. 5 | Process maps generated by the video vision transformer model.** The IN718 (**a**) and Ti-6Al-4V (**b**) maps are obtained by the model trained on SS316L, whereas the SS316L map (**c**) is obtained by the model trained on IN718. The percentages of correctly identified classes are plotted above the P-V datapoints. The misclassified data point in the Ti-6Al-4V map is misidentified as desirable. Source data are provided as a Source Data file.

**Table 3 | Comparison of the classification accuracies (%) obtained by state-of-the-art convolutional neural network (CNN) and transformer-based models**

| Training on | Testing on | VGG16 | ResNET152 | MovieNET-A3 | TimeSformer | ViViT-B |
|---|---|---|---|---|---|---|
| Ti-6Al-4V | SS316L | 83.30 | 87.48 | 81.96 | 88.80 | 90.24 |
| | IN718 | 82.50 | 90.72 | 82.99 | 92.36 | 94.48 |
| SS316L | Ti-6Al-4V | 86.70 | 84.99 | 88.56 | 90.52 | 90.22 |
| | IN718 | 97.37 | 96.36 | 98.26 | 97.38 | 98.04 |
| IN718 | Ti-6Al-4V | 88.37 | 90.12 | 93.67 | 95.54 | 94.36 |
| | SS316L | 89.39 | 93.22 | 92.06 | 90.80 | 92.48 |

The mean values of the accuracies are reported and averaged over five independent training runs.

models. The pretrained ViViT-B model has a more balanced performance between the test datasets and achieves a classification accuracy higher than 90%. The model configurations and training details can be found in the Supplementary Method section.

To provide more insights into the method performance, we visualize the t-SNE projection[51], i.e., a technique used for the visualization of high data dimensions, of the ViViT-B model features in Fig. 9a. The figure illustrates separable melt-pool classes. The attention map[52] in Fig. 9b illustrates the mean attention weight over the transformer heads after linear scaling. The map shows that the dynamic features of the melt pool at the keyhole and the tail carry more attention.

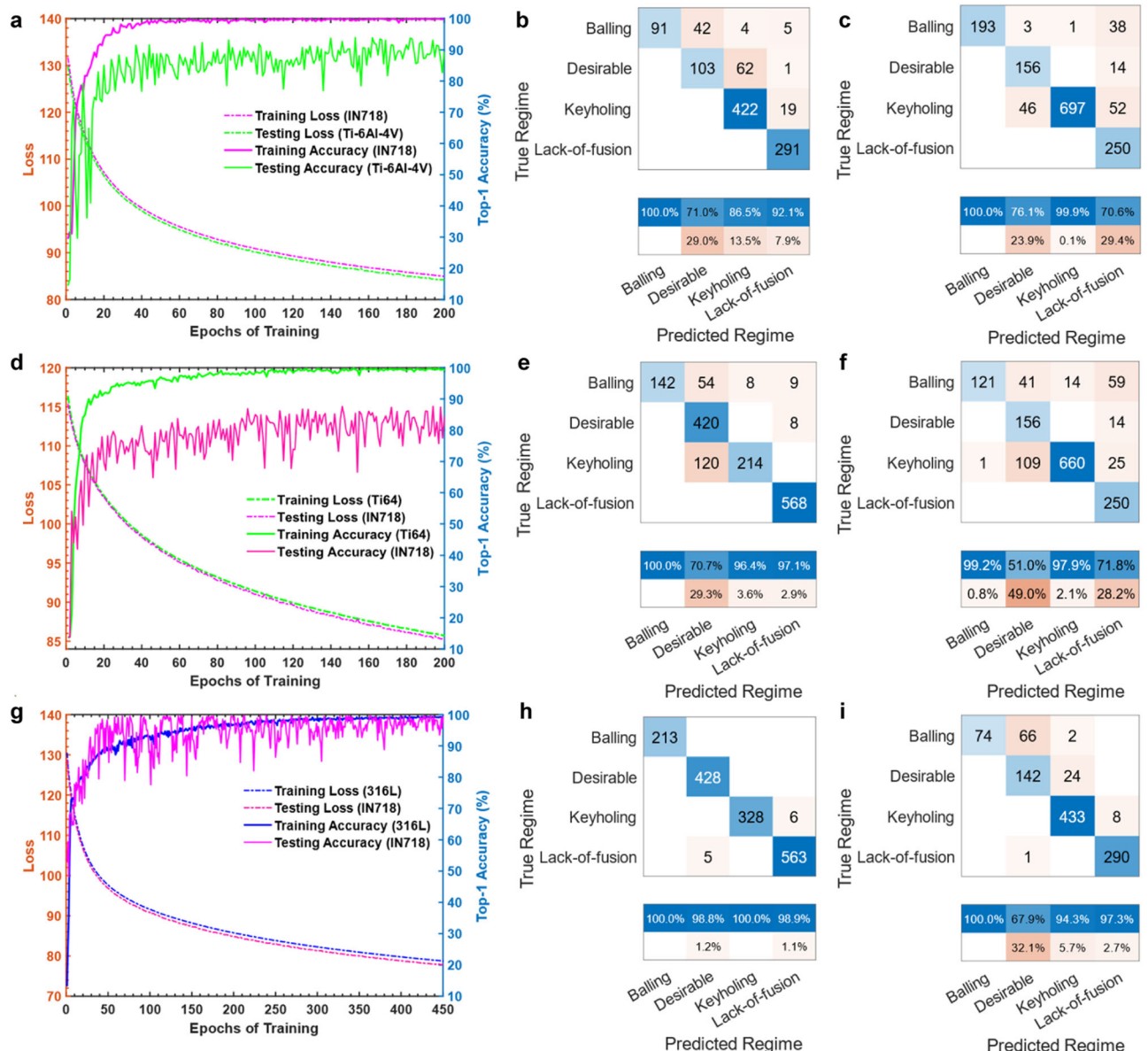

**Fig. 6 | Results of the cross-dataset evaluations.** Results by training on IN718: (**a**) training history, (**b**) confusion matrix of Ti-6Al-4V, (**c**) confusion matrix of SS316L. Results by training on Ti-6Al-4V: **d** training history, (**e**) confusion matrix of IN718, (**f**) confusion matrix of SS316L. Results by training on SS316L: (**g**) training history, (**h**) confusion matrix of IN718, (**i**) confusion matrix of Ti-6Al-4V. Source data are provided as a Source Data file.

The results of three model variants of ViViT-B[21] are shown in Supplementary Table S5. The three models are different in their attention patterns and how the spatiotemporal information flows. The spatiotemporal attention model has one transformer, the factorized encoder model has two separate transformer encoders for spatial and temporal tokens, and the factorized self-attention model has one transformer as the first model, but spatial self-attention is computed first followed by temporal self-attention[21]. The factorized encoder model shows superior performance on most of the test datasets, especially when trained on the Ti-6Al-4V alloy dataset, which is observed to be more difficult for training, as shown above.

The influence of regularization is examined by running experiments with the factorized encoder model of ViViT-B with and without data augmentation steps of random horizontal flipping, random cropping and stochastic layer dropout[53]. The results are shown in Supplementary Table S6. The results highlight the importance of regularization in training, as the performance is improved, especially in the Ti-6Al-4V experiments.

## Variability mapping

Variability in melt-pool morphologies can lead to dimensional inconsistency of printed features and a coarse surface finish. Moreover, it can indicate heat accumulation and fusion defects, such as a discontinuities in melt tracks and spatter generation and powder spreading defects. Therefore, selecting process parameters that lead to a less variable melt pool can enhance the consistency of the printing outcome.

Following the creation of process maps for printability and defect formation, we construct process maps for morphological variability. Although the depth of the melt pool cannot be seen from the top view of the machine bed, the width and area attributes are markers of printing consistency[54].

Powder single tracks are printed in Ti-6Al-4V alloy with a layer thickness of 30 μm, a 100-μm laser spot diameter, and different processing parameter combinations: laser powers of 200, 300, and 400 W and scanning speeds of 800, 1200, and 1600 mm/s. The experiments are conducted four times to account for variability in powder

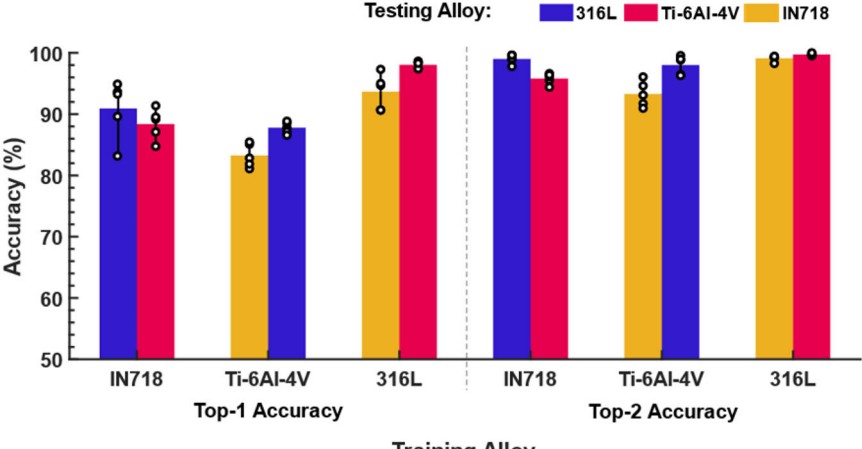

**Fig. 7 | Results of independent runs with reshuffled datasets.** Average percentages of the top-1 and top-2 classification accuracies. The accuracy values are averaged across five independent training runs with different random seeds and data shuffling. The training datasets are randomly shuffled each run. The results of all runs are illustrated with circles. Source data are provided as a Source Data file.

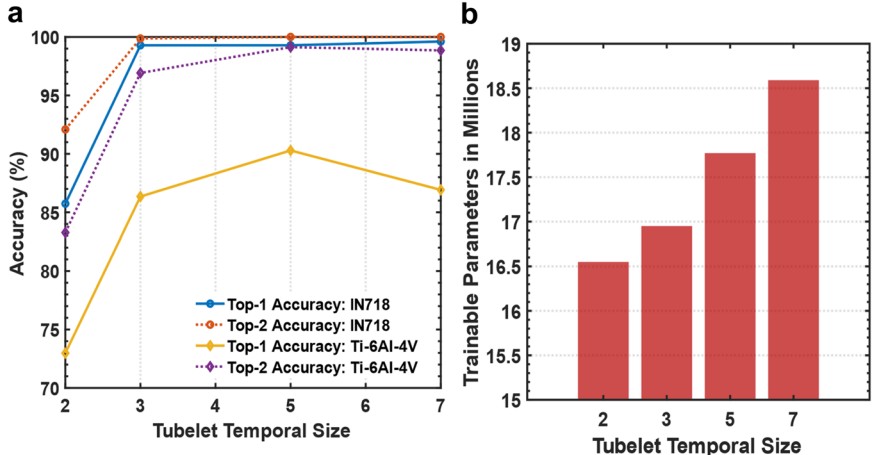

**Fig. 8 | Ablation analysis.** Effect of varying the backbone capacity on the prediction accuracy (**a**) and the number of trainable parameters (**b**). Source data are provided as a Source Data file.

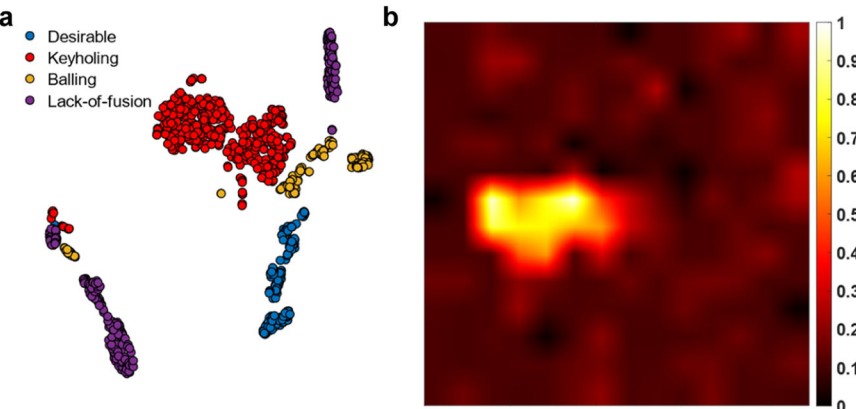

**Fig. 9 | Analysis of the transformer model. a** t-SNE[51] of the ViViT-B features shows separable melt pool classes. **b** attention map[52] shows the mean attention weight over the transformer heads after linear scaling. The map illustrates that the dynamic features of the melt pool at the keyhole and the tail carry more attention. Source data are provided as a Source Data file.

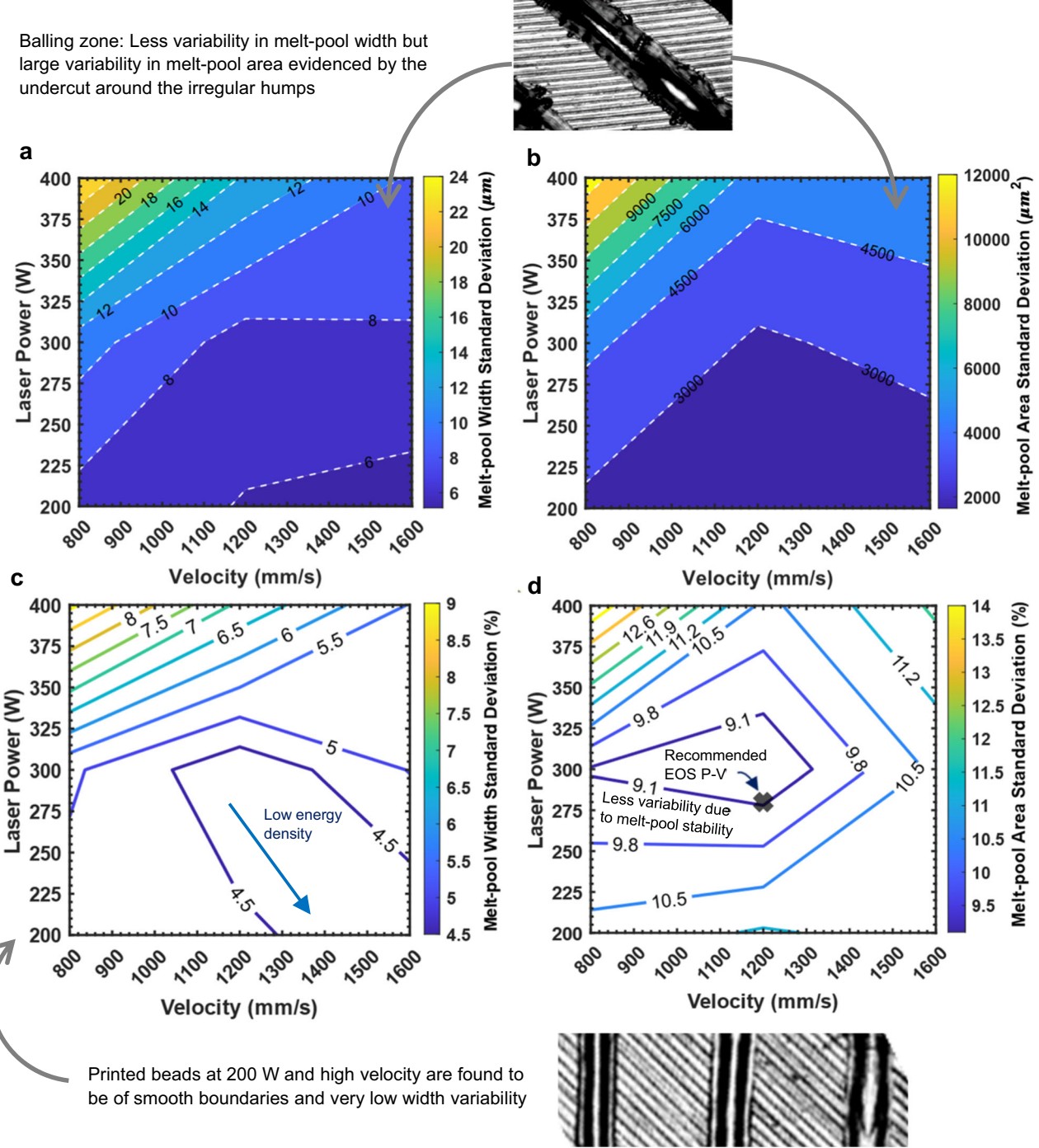

**Fig. 10 | Process maps of melt pool morphological variability.** The morphological variability of Ti-6Al-4V alloy represented by the standard deviation of the width and area of the melt pools captured by high-speed imaging at 54000 frames per second. **a** Standard deviation of melt-pool width in μm. **b** Standard deviation of melt-pool area in μm². **c** Relative standard deviation of melt-pool width. **d** Relative standard deviations of melt-pool area. Relative standard deviations are calculated as the percentage of the standard deviation values to the arithmetic mean values. Source data are provided as a Source Data file.

spreading and to sufficiently analyze melt-pool changes. After printing the beads and collecting the data, ex situ analysis with a ZEISS Axio Imager microscope was performed to measure the track width every 200 μm. The liquidus-solidus threshold in the captured videos is estimated by correlating the actual bead's mean width with the melt pool captured by the camera.

The variability in melt-pool morphology represented by the standard deviation of the melt-pool width and area is illustrated in Fig. 10. As shown in the figure, a large standard deviation is observed

at high energy densities, where the melt pools are large and contract as the energy density decreases. The percentage of the standard deviation (Fig. 10b) also generally follows the same trend. Although the width variability is not high, the variation in the melt-pool area is significant. An explanation for this is depicted in Fig. 10. As observed by the microscope, beads with balling can have smooth boundaries, as the undercuts occupy the areas around the irregular humping features at the center of the beads. Beads printed at 200 W and high velocities are found to have smooth boundaries and small standard

deviations in the width. This finding is the same as the ex-situ observations of the printed beads. An interesting finding is that the recommended P-V combination by the machine manufacturer is at the lowest area variability of the melt pool, which is an indication of melt-pool stability.

## Discussion

As demonstrated above, molten pool dynamics and shape changes can be captured with a simple off-axial imaging setup. Vision transformers with temporal embedding can enable in situ detection of melt-pool defects and efficient process mapping for alloys that are different in chemical composition and intrinsic thermofluid properties from the material used in training. The process maps generated by our method, along with the variability maps of molten pool attributes, can potentially accelerate the qualification of printability and process development for newly developed 3D printed alloys.

We show that incorporating the temporal features leads to accurate prediction of process maps, as the dynamics of the process are a major contributor to defect formation and the printing regime[20,30]. Consistent with former observations[15,54], balling morphology and keyholing porosities may not be present in all frames or vertical cross-sections of the printed beads. The balling defect is known to be periodic, which strengthens our argument that individual images are not sufficient to infer the printing regime of the processing parameters. Moreover, clear melt-pool shapes and indications for the dynamics of defect formation can be captured in situ by a simple off-axial monitoring setup without the need for calibration to obtain temperature measurements or modification of existing printing machines. The pure transformer model with layer weight regularization is found to be robust and generalizable to different metal alloy datasets.

Results obtained from training on the Ti-6Al-4V data tend to be inferior to those obtained from training on the other alloy datasets, as the thermophysical properties of this metal alloy are significantly different from those of the other alloys[41]. Pretrained ViViT with data augmentation results in an improved performance in the Ti-6Al-4V experiments due to improved initialization and reduced variance.

The classification accuracies of the model are found to be high enough to reconstruct the process maps of the testing alloys. However, the "keyholing" defect class is confused with the "desirable" class in some instances. The desirable class consists of data points of printed beads in conduction and keyholing regimes, as keyholing without severe evaporation may not cause defects. The findings of this study can be the basis for developing deep learning models with multilabel classification.

Although the generalizability of the method is validated, its expansion to real-time monitoring for process control has some challenges and limitations, and further studies are needed to address these limitations. First, the data transfer rate must be high enough to match the recording rate needed to capture the high frequency oscillations in melt-pool changes. Second, most commercial high-speed cameras are designed to record short videos for limited periods.

## Methods

### Experimental setup

The high-speed imaging setup was built at the Mill 19 facility of the CMU Next Manufacturing Center (Pittsburgh, PA). A Photron FASTCAM Mini AX200 high-speed monochrome camera (Tokyo, Japan) was used to capture the melt pool in real time at 54,000 frames per second. The exposure time of the camera was manually adjusted depending on the brightness level of the scene to avoid saturation and blooming, i.e., exceeding the electric signal limit that the camera sensor can handle[55]. The experiments were conducted with a TRUMPF TruPrint 3000 laser powder bed machine (Farmington, CT). Argon gas was used with continuous circulation (an oxygen

concentration of less than 0.1%) to mitigate the plume intensity above the melt pool. Small plates were used for the bare plate experiments on Ti-6AL-4V, IN718, and SS316L from McMaster-Carr (Elmhurst, IL). Gas-atomized Ti-6AL-4V powder (EOS Titanium Ti64 Grade 23 powder from EOS GmbH, Germany) was used in the variability study. The chemical compositions of the materials can be found in Supplementary Tables S7–S10. The numbers of samples, i.e., data points, per track were different in different experiments, as the number depends on the scanning velocity. A total of 1340 tracks were printed for the three alloys used in the study. Each track was 6 mm long, and 1.5–2 mm of the track ends were not captured. The samples were sectioned using wire electrical discharge machining and polished and etched according to the ASTM E407 standard. The ex situ analyses were performed using a ZEISS Axio Imager microscope (Oberkochen, Germany) for ground truth labeling and measuring the width of the printed tracks.

### Data processing

The recorded videos were processed using scaling and image registration. The frames were converted into grayscale arrays, and the molten pools were registered to be in the same location along each clip. The image intensity was normalized. For each video, the pixel intensities were linearly scaled from zero to one, where one represented the maximum pixel intensity in the video. The size of each data point was $80 \times 160 \times 15$, which represents a temporal resolution of 18.5 µs and a spatial resolution of 6.3 µm. Active contouring followed by thresholding was used to measure the melt-pool attributes for the attribute variability analysis.

Ground truth labeling was performed based on four classes: (1) keyholing defects: when the keyhole penetrates deep enough into the material with a probability to generate pores (the width to depth ratio is less than $1.2$[9,56] or if keyholing porosities are observed); (2) balling: any track that exhibits peaks with a ball-like shape is classified as balling even if the track is shallow; (3) lack-of-fusion: when the printed track is very shallow and no balling is observed; and (4) desirable: if the track does not meet the criteria of the other classes. To ensure the desirable printing class is free of possible defects, any defect morphology observed in one of the printed tracks is sufficient to label the P-V combination as belonging to this defect class.

### Deep learning model

We use the video vision transformer ViViT[21], which is a pure transformer model with tubelet embedding of the video clips (i.e., feeding the model with nonoverlapping spatiotemporal information[21]). The extracted information from the input contains both spatial and temporal information of the melt pool as the laser travels. The spatiotemporal attention model is used with the transformer encoder[33], which includes multihead self-attention and an MLP layer with layer normalization and residual connections. The architecture of the model is illustrated in Fig. 1. The model has 12 transformer heads and 20 layers, and the MLP hidden dimension is 256. To mitigate overfitting, we use the elastic net method[57], applying L1 and L2 weight regularization to the MLP layers. The calculations are performed in TensorFlow. The transformer model was trained on a single NVIDIA Tesla V100 DGXS GPU with 32 GB. The number of training epochs ranges between 200 and 500 depending on convergence, and the batch size is set to 128. The configurations of the benchmarking models can be found in the Supplementary Method section.

## Data availability

Data to support the findings and conclusions are included in the paper and the Supplementary Information. Samples of the recorded videos are included in the Supplementary Movies. Other datasets are available upon request from the corresponding author. Source data are provided with this paper.

## Code availability

The computer codes developed for this study are available from the corresponding author upon request.

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

## Acknowledgements

We acknowledge the funding support by the Army Research Laboratory (W911NF-20-2-0175 by J.B.) and the U.S. NAVY SBIR (N162-083 by J.B.). The views and conclusions contained in this document are those of the authors and should not be interpreted as representing the official policies, either expressed or implied, of the Army Research Laboratory, the U.S. NAVY, or the U.S. Government. In addition, we acknowledge the use of the Materials Characterization Facility at Carnegie Mellon University, USA supported by grant MCF-677785.

## Author contributions

D.G., C.T. and J.B. conceptualized the project. D.G. designed and performed the experiments; D.G. performed the computational work and programming. D.G. performed the characterization and data analysis. D.G., C.T. and J.B. analyzed the results. J.B. and C.T. supervised the project. J.B. conceived the funding. D.G. wrote the manuscript with input from C.T., and J.B. All authors read the manuscript and commented on the paper.

## Competing interests

D.G., C.T. and J.B. declare that an international patent application has been filed by Carnegie Mellon on the proposed method of accelerating process development, PCT/US2323/078686. The authors have no other competing interests to declare.
