## [Peer Review File · Nature Communications]

Accelerating Process Development for 3D Printing of New Metal AlloysREVIEWER COMMENTS

Reviewer #1 (Remarks to the Author):

The authors address an important topic in metal additive manufacturing, namely speeding up process parameter development for new alloys. They use in-situ monitoring data in combination with a machine learning approach to create process maps for different alloys. Although there are some encouraging results, I think there are some major weaknesses in the paper.

Strengths:

- The model appears to work quite well for predicting the different melting regimes between different materials. The model converges suggesting the high-speed imaging is picking up similar signals from each regime between the different materials.
- Though the model is simple to implement, and is fairly off the shelf based on the details (there is a Keras tutorial with code to implement it), it hasn't been used before for this specific problem to my knowledge.

Major comments:

- It is unclear why the model can generalize between the datasets. The authors state in line 7/8 p3 that the optical train is devised to block plume emissions. How was this achieved? Did this need to be adjusted between experiments to account for different melting temperatures or was the data collection method the same? Was there any normalisation of the image intensity statistics required in data processing? Line 24/24 p2: "We devised an approach to get rid of the saturation in captured frames..." but this is not explained. As it stands the study would be hard to replicate.
- Fig. 4 G, H, I should be separated to indicate this was the original training/testing model. I suspect the model was trained on 316 and validated (the authors call it testing) on IN718. That is why the model does so good there. The authors do not look at Ti64 until after this. Why did Ti64 underperform? There is an explanation in line 12-14 p5 but no data to back up the claim. The model appears to be overfitting to IN718. Also, it is not clear if hyperparameters were changed when performing this validation? It would also be interesting to see how sensitive the results are to a change in random seed.
- The model required heavy regularization to work, suggesting that the model is too complex for the problem at hand, not just that there is a small amount of data. There are no comparisons to simpler models. e.g. a smaller CNN, CNN + LSTM or VRNN or using melt pool features plus a dynamics feature such as the image correlation from Fig3B and/or standard deviation of area from Fig. 8 and possible temporally smoothing the data with e.g. Logistic Regression, KNN or RF type model. This would demonstrate the performance of the method.
- There is no ablation study to determine importance of dynamics, e.g. claim of line 9, p7 that there is additional performance gain from including dynamics can't be verified. Fig.3B&C seems to show keyhole is most important when it comes to temporal changes but is not quantified across the dataset, only one material is shown. A statistical analysis could have been done to quantify this. Fig. 7 is on the right track to answering this question, but it is not clear whether having more model parameters or increasing the temporal context was the important variable. The figure is not well described in the text (line 3-8 p6).
- Details of the architecture are not given, e.g. how many transformer blocks, how many hidden dimensions, details of the MLP, etc. Also, Transformers are typically pretrained, was a pretrained model used?
- It is not clear what is meant by "sample" in line 15 p5. Does it mean one image frame? is it the same as sample in Table 1 and 2? Or is one track one sample? How many tracks were printed? Line 10, p8 seems to show a data point had 15 frames, is this the same as one sample? Then how many samples were there per track? How many tracks? What was the track length?
- Line 29 p8, the equation describes a regression loss yet the paper describes a classification problem. Why not write the one used instead of the equation for Elastic net? Also, this type of regularization is usually applied to the model weights not the model output. Line 26 p8 "to overcome overfitting ...

applying L1 and L2 weight regularization to the MLP layers' output."

- Line 2, p3 states: "we generate variability process maps for molten pool attributes to guide for determination 2 of optimal hatch spacing" but this is not revisited. Is this what Fig. 8 is referring to? Also Fig 8. shouldn't the top and bottom row have the same contours, maybe some labels/titles are missing or am I missing something?
- There is a lack of detailed interpretation of the results. For example, keyhole oscillations are observed but this not a new insight and nor is it well quantified here.

Minor comments:

- Line 1, p2 - variability problem not defined, no references.
- Fig.2 - scales are different for each image, this seems sloppy and is even mentioned in the caption!
- There are some spelling mistakes and incorrect figure references e.g. line 5 p4, line 27 p7
- Fig.4 - these are confusion matrices not cross-correlation
- Line 6-8 p6 doesn't seem like a valid claim. The melt pool shape/temperature is also different, not just dynamics. It is not the same object.
- Fig.7 - it is not well defined what Tubelet temporal size means for an average reader unfamiliar with this model. Also "tublet" misspelled in axis labels.

Reviewer #2 (Remarks to the Author):

This paper proposed the use of video vision transformers trained on captured frames by the high-speed camera to monitor the dynamic changes of the molten pool and use the temporal data to classify the process into different types of defects.

The major benefits are (1) Achieving high algorithmic accuracy to over while sparing the use of high-cost pyrometers to extract the temperature field and (2) Generalizability to new alloys.

While the paper is well-motivated and easy to follow, I have two major questions:

(1) Why vision transformers (ViTs)? The paper argues ViTs are essential for the proposed task. However, there are no ablation studies and comparisons to standard convolutional neural networks (CNNs), such as residual networks. It's known that in principle, ViTs require more data for training/pre-training compared to CNNs, and the results may not be better. Moreover, there are many CNN-based networks optimized for temporal-spatial data (like video). So it is unclear, based on the current results, about the role and justification of the use of ViTs in this context.

(2) Vague connection to 3D printing. I am not an expert in 3D printing. But most results I've seen in this paper are based on digital data or some statistics. Did the authors actually implement the proposed approach for 3D printing? If not, is there a direct connection between the presented result to 3D printing?

Response Letter

We highly appreciate the feedback and comments we received from the reviewers. The changes in the manuscript file are highlighted in yellow. We have answered the reviewers' questions and addressed their concerns. The manuscript has been revised accordingly. Our responses to the comments are provided below.

Reviewer #1:

The authors address an important topic in metal additive manufacturing, namely speeding up process parameter development for new alloys. They use in-situ monitoring data in combination with a machine learning approach to create process maps for different alloys. Although there are some encouraging results, I think there are some major weaknesses in the paper.

Strengths:

- The model appears to work quite well for predicting the different melting regimes between different materials. The model converges suggesting the high-speed imaging is picking up similar signals from each regime between the different materials.
- Though the model is simple to implement, and is fairly off the shelf based on the details (there is a Keras tutorial with code to implement it), it hasn't been used before for this specific problem to my knowledge.

Major comments:

1. It is unclear why the model can generalize between the datasets. The authors state in line 7/8 p3 that the optical train is devised to block plume emissions. How was this achieved? Did this need to be adjusted between experiments to account for different melting temperatures or was the data collection method the same? Was there any normalisation of the image intensity statistics required in data processing? Line 24/24 p2: "We devised an approach to get rid of the saturation in captured frames..." but this is not explained. As it stands the study would be hard to replicate.

Response to comment 1:

We appreciate the encouraging comments and feedback.

Why can the model generalize between the datasets?

There are common patterns in printing regimes that can be captured. The topology of the melt pool and how the shape dynamically changes are two features that are utilized in our method in comparison to other approaches. Although there is no simple answer to this question as the deep learning model does not use hand-crafted features. In the revised manuscript, we have included an attention map that shows what are the features in the melt pool that carry more attention weight. Additionally, we have added t-SNE plot (See Fig. 10) that shows the model features are more separable using the proposed method. Please, see Page 6, lines 25-29.

How was this achieved?

Minimization of the plume that can cover the melt pool and mitigation of the saturation in captured videos are two major requirements to take advantage of the melt pool dynamic features in the input data. There are multiple ways to minimize the blocking plume on the melt pool images. The setup differs from one machine to another and depends on the imaging setup. One approach is to keep the oxygen concentration in the chamber to a minimum. For instance, Bidare et al. [1] showed how a high shield gas flow rate can remove away the plume. Filtering the wavelength that is associated with the plume and plasma emission is another possibility. We have used these approaches to make the melt pool as clear

as possible. As long as the melt pool is not blocked and the topological features are observable in the captured videos, these features can be utilized, making the method repeatable and reproducible. Please, see the added note on Page 9, lines 5-7, 9-10.

Did this need to be adjusted between experiments to account for different melting temperatures or was the data collection method the same?

The experimental setup and data collection method are kept unchanged throughout the experiments. The only parameter that needs to be changed is the exposure time of the imaging sensor. It is manually adjusted during imaging to avoid saturation in captured frames as we pointed out in the response above (see Page 9, lines 5-7).

Was there any normalisation of the image intensity statistics required in data processing?

The image intensity is normalized in the data processing in order to eliminate the influence of that parameter. For each video, the pixel intensities are linearly scaled from zero to one, where one represents the maximum pixel intensity in the video. We clarified that on Page 9, lines 24-25.

2. Fig. 4 G, H, I should be separated to indicate this was the original training/testing model. I suspect the model was trained on 316 and validated (the authors call it testing) on IN718. That is why the model does so good there. The authors do not look at Ti64 until after this. Why did Ti64 underperform? There is an explanation in line 12-14 p5 but no data to back up the claim. The model appears to be overfitting to IN718. Also, it is not clear if hyperparameters were changed when performing this validation? It would also be interesting to see how sensitive the results are to a change in random seed.

Response to comment 2:

Figure 4 shows the results of three separate experiments for the cross-dataset evaluation. All hyperparameters are kept the same throughout all the experiments. Only the datasets are changed. In each experiment, the model is trained on one alloy and tested on the two other alloys. The history of one of the test datasets is illustrated in the left column for over/underfitting evaluation purposes. The validation/test datasets were not seen by the model during training and their performance was not utilized to enhance the performance.

Why did Ti64 underperform?

It is possible that the model performs best on one alloy and underperforms on another. However, we have shown in the added experiments on page 6, that by using pre-trained ViViT-B with data augmentation the performance is more balanced as the network regularization is improved.

In our experiments, the Ti-6Al-4V alloy is observed to emit a denser vapor plume in comparison to the other alloys. Ti-6Al-4V has significant vaporization of its alloy elements in comparison to 316L and IN718 [2]. Moreover, Ti-6Al-4V has lower thermal conductivity and higher absorptivity to laser radiation, and much different thermo-physical properties [3]. Thus, the classification accuracy of Ti-6Al-4V is expected to be lower than that of IN718. We added our hypothetical explanation on Page 5, lines 22-25.

it is not clear if hyperparameters were changed when performing this validation?

All hyperparameters are kept the same throughout all the experiments.

It would also be interesting to see how sensitive the results are to a change in random seed.

In order to test the influence of random initiation, the experiments are repeated five times with a different random seed, and training data shuffling is performed each run. Bar plots of the Top-1 accuracies are shown in Fig. 7. The experiments show a significant influence of the initiation as the model is trained from scratch and the training data is reshuffled in each experiment (See Page 6, lines 5-10).

3. The model required heavy regularization to work, suggesting that the model is too complex for the problem at hand, not just that there is a small amount of data. There are no comparisons to simpler models. e.g. a smaller CNN, CNN + LSTM or VRNN or using melt pool features plus a dynamics feature such as the image correlation from Fig3B and/or standard deviation of area from Fig. 8 and possible temporally smoothing the data with e.g. Logistic Regression, KNN or RF type model. This would demonstrate the performance of the method.

Response to comment 3:

Regularization is strongly recommended for training from scratch [4]. We have performed experiments using a pre-trained ViViT-B model without regularization for only 5 epochs and the same model with layer dropout (probability of 0.1) and augmentation (random image crop and flip). The performance was comparable in some experiments as in (train: 316L, test: IN718) and (train: IN718, test: 316L) to 8.94% less efficient in the case of (train: IN718, test: Ti-6Al-4V). Please, see Supplementary Table S6.

In order to validate the performance of the method, experiments are performed to compare the video vision transformer with other state-of-the-art models. Two pre-trained video vision transformer models ViViT-B [5] and TimeSformer [6] are compared with Deep Convolutional Network (VGG16) [7], Deep Residual model (ResNet152) [8], and Mobile Video Network model (MoViNet-A1) [9]. As the results in Table 3 show, the video vision transformer models outperform the CNN-based models. The pre-trained ViViT-B model has a more balanced performance between the test datasets, achieving Top-1 accuracy higher than 90%. The training details in addition to more comparisons and ablation studies for the model variants can be found in the Supplementary Method.

It is possible to use traditional computer vision approaches or handcrafted features for printing regime classification. However, there are two aims we target: (1) Generalizability to new alloys: instead of relying on material-specific features (which may not be known especially when printing with new alloys), use only raw high-definition videos of the molten pools (the dynamics of the molten pool are obviously different in each printing regime, and their behavior is quite similar across the materials we tested so far). (2) Affordability: the data collection method should be affordable to most additive manufacturing facilities.

Training an ML model on a specific alloy can easily be overfitted on its dataset and perform poorly on a different material, and hence typically needs fine-tuning. For instance, a complex setup and installation process is needed for temperature field measurement [10, 11], which may require imaging alignment, calibration, and a special setup to be integrated with the scanning head of the 3D printing machine. In addition, melt pool attributes are material-specific. Thus, incorporating these handcrafted features can be more suitable for the real-time detection of printing defects of alloys with known characteristics. An example of existing methods can be found in [12]. Please, see the highlighted changes on Page 6, lines 17-35, Page 7, lines 1-7, and Table 3.

4. There is no ablation study to determine importance of dynamics, e.g. claim of line 9, p7 that there is additional performance gain from including dynamics can't be verified. Fig.3B&C seems to show keyhole is most important when it comes to temporal changes but is not quantified across the dataset, only one material is shown. A statistical analysis could have been done to quantify this. Fig. 7 is on the right track to answering this question, but it is not clear whether having more model parameters or increasing the temporal context was the important variable. The figure is not well described in the text (line 3-8 p6).

Response to comment 4:

We have benchmarked the ViViT model with other deep-learning models that use only 2D images. The classification accuracy was up to 4.21% higher than that of the 2D CNN model, ResNet152. In addition, we have shown by attention maps that the model weights are higher on key dynamic attributes of the melt pool such as the keyhole. We have performed statistical analyses across the other alloys and added them to Fig. 3.

The only variable in this ablation study is the number of frames in each input tube. That increases the number of parameters, however, as also observed in [5], larger models can boost the performance. In this experiment, the model size is kept unchanged. We have added a detailed description to the text.

5. Details of the architecture are not given, e.g. how many transformer blocks, how many hidden dimensions, details of the MLP, etc. Also, Transformers are typically pretrained, was a pretrained model used?

Response to comment 5:

We have added a new section, Supplementary Method, which includes details on the models' configurations and training. In addition, we have performed more experiments with pre-trained models ViViT-B [5] and TransFormer [6]. Please, see the highlighted changes on Page 6, lines 17-35, Page 7, lines 1-7, Table 3, and the Supplementary Method.

6. It is not clear what is meant by “sample” in line 15 p5. Does it mean one image frame? is it the same as sample in Table 1 and 2? Or is one track one sample? How many tracks were printed? Line 10, p8 seems to show a data point had 15 frames, is this the same as one sample? Then how many samples were there per track? How many tracks? What was the track length?

Response to comment 6:

We meant “video” by “sample” in line 15 p5. Each sample or data point is a video of 15 frames. We have clarified the definition in the captions of the tables. One track can be multiple samples (i.e., data points). The number of samples per track is different from one to another as it depends on the scanning velocity. In total, 1340 tracks were printed for the three alloys used in the study. Each track is 6mm long; 1.5-2mm of the track’s ends are not used in the datasets to avoid transient behavior. Please, see the highlighted changes on Page 9, lines 14-17.

7. Line 29 p8, the equation describes a regression loss yet the paper describes a classification problem. Why not write the one used instead of the equation for Elastic net? Also, this type of regularization is usually applied to the model weights not the model output. Line 26 p8 “to overcome overfitting ... applying L1 and L2 weight regularization to the MLP layers’ output.”

Response to comment 7:

That is true. It is weight-based regularization. We have improved the writing for convenience. Now it reads: “To mitigate overfitting, we use the elastic net method, applying L1 and L2 weight regularization to the MLP layers.” It is used for all layers not only the output, so we have removed the equation to avoid confusion. Please, see the highlighted changes in Page 10, lines 8-10.

8. Line 2, p3 states: “we generate variability process maps for molten pool attributes to guide for determination 2 of optimal hatch spacing” but this is not revisited. Is this what Fig. 8 is referring to? Also Fig 8. shouldn't the top and bottom row have the same contours, maybe some labels/titles are missing or am I missing something?

Response to comment 8:

The melt pool attributes can be used for the optimization of hatch spacing. We have illustrated it as a part of a framework for in-situ process development. However, we did not include the methodology as it is beyond the topic of the current study. Fig. 9 (top- now “A”) illustrates the standard deviation in mm while the (bottom -now “B”) shows the standard deviation as a percentage. We have added details for clarification in the figure's caption.

9. There is a lack of detailed interpretation of the results. For example, keyhole oscillations are observed but this not a new insight and nor is it well quantified here.

Response to comment 9:

The section of the discussion has been expanded to include a detailed interpretation of the results.

The keyhole behavior and oscillations are well-studied in the literature [13, 14]. Although the oscillations are observable in our results and the topologies of the melt pools are relatively clear, their quantifications for concluding results require a different design for the experimental setup. The text is enhanced with references that quantified keyhole fluctuations. Please, see Page 3, lines 26-33 – Page 4, lines 1-2.

Minor comments:

1. Line 1, p2 - variability problem not defined, no references.
2. Fig.2 - scales are different for each image, this seems sloppy and is even mentioned in the caption!
3. There are some spelling mistakes and incorrect figure references e.g. line 5 p4, line 27 p7
4. Fig.4 - these are confusion matrices not cross-correlation
5. Line 6-8 p6 doesn't seem like a valid claim. The melt pool shape/temperature is also different, not just dynamics. It is not the same object.
6. Fig.7 - it is not well defined what Tubelet temporal size means for an average reader unfamiliar with this model. Also "tubelet" misspelled in axis labels.

Response minor comments:

Thank you for pointing out to these mistakes, We have corrected them in the revised version.

1. Line 1, p2 - variability problem is defined, and references are added.
2. Fig.2 – We have changed the scale of each image to make it easier to read.
3. We have corrected the typos and the spelling mistakes.
4. Fig.4 – We have corrected this typo.
5. Line 6-8 p6 – In other classification problems, distinct objects such as cats, dogs, and cars are classified, but in the problem at hand there is one object which is melt pool with different characteristics. We agree that the melt pool dynamics is not the only utilized feature. We have updated this statement to be more accurate. Please, note however that temperature is not utilized in our method.
It is revised to *"In this particular classification problem, the model distinguishes the same object but with different topological shapes and dynamics"*. (Now p6, lines 13-15)
6. Fig.7 – Thanks for pointing out this typo. We have corrected it. Although we have defined it in the Materials and Methods section, we have changed it to "input tube" for better readability.

Reviewer #2 (Remarks to the Author):

This paper proposed the use of video vision transformers trained on captured frames by the high-speed camera to monitor the dynamic changes of the molten pool and use the temporal data to classify the process into different types of defects.

The major benefits are (1) Achieving high algorithmic accuracy to over while sparing the use of high-cost pyrometers to extract the temperature field and (2) Generalizability to new alloys.

While the paper is well-motivated and easy to follow, I have two major questions:

- (1) Why vision transformers (ViTs)? The paper argues ViTs are essential for the proposed task. However, there are no ablation studies and comparisons to standard convolutional neural networks (CNNs), such as residual networks. It's known that in principle, ViTs require more data for training/pre-training compared to CNNs, and the results may not be better. Moreover, there are many CNN-based networks optimized for temporal-spatial data (like video). So it is unclear, based on the current results, about the role and justification of the use of ViTs in this context.

Response to comment 1:

Thank you for pointing out this. The ultimate objective to make ML effectively used for the process development of new alloys is to be more generalizable to unseen data with fewer assumptions about the data distribution. ViTs have less image-specific inductive bias than CNNs [4]. In addition, the way of the input embeddings and slicing of the videos into nonoverlapping patches without embedding their positions can make it more suitable for the problem at hand.

To validate the performance of the method, experiments are performed to compare the video vision transformer with other state-of-the-art models. Two pre-trained video vision transformer models ViViT-B [5] and TimeSformer [6] are compared with Deep Convolutional Network (VGG16) [7], Deep Residual model (ResNet152) [8], and Mobile Video Network model (MoViNet-A1) [9]. As the results in Table 3 show, the video vision transformer models outperform the CNN-based models. The pre-trained ViViT-B model has a more balanced performance between the test datasets, achieving Top-1 accuracy higher than 90%. The training details in addition to more comparisons and ablation studies for the model variants can be found in the Supplementary Method. Please, see the highlighted changes on Page 6, lines 17-24, and Table 3.

(2) Vague connection to 3D printing. I am not an expert in 3D printing. But most results I've seen in this paper are based on digital data or some statistics. Did the authors actually implement the proposed approach for 3D printing? If not, is there a direct connection between the presented result to 3D printing?

Response to comment 2:

The datasets we have collected are based on captured videos of 3D-printed single beads on a laser powder bed machine, TRUMPF TruPrint 3000, at the Mill 19 facility of the CMU Next Manufacturing Center (Pittsburgh, PA). We have included a new figure, Fig. 4 (see below) in the revised manuscript that illustrates the 3D printing defects in more detail.

Figure R1. Micrographs of sectioned beads and top views of printed single beads that depict the four printing classes: (A) lack-of-fusion, (B) desirable, (C) balling, and (D) keyholing.

References

1. Bidare P, Bitharas I, Ward RM, Attallah MM, Moore AJ (2018) Fluid and particle dynamics in laser powder bed fusion. *Acta Mater* 142:107–120
2. Mukherjee T, Zuback JS, De A, DebRoy T (2016) Printability of alloys for additive manufacturing. *Sci Rep* 6:1–8
3. Mills KC (2002) Recommended values of thermophysical properties for selected commercial alloys. Woodhead
4. Dosovitskiy A, Beyer L, Kolesnikov A, et al (2020) An Image is Worth 16x16 Words: Transformers for Image Recognition at Scale. <https://doi.org/10.48550/arxiv.2010.11929>
5. Anurag A, Dehghani M, Heigold G, Sun C, Lučić M, Schmid C (2021) ViViT: A Video Vision Transformer. 2021 IEEE/CVF International Conference on Computer Vision (ICCV)
6. Bertasius G, Wang H, Torresani L (2021) Is Space-Time Attention All You Need for Video Understanding? *Proc Mach Learn Res* 139:813–824
7. Simonyan K, Zisserman A (2015) Very deep convolutional networks for large-scale image recognition. 3rd International Conference on Learning Representations, ICLR 2015 - Conference Track Proceedings
8. He K, Zhang X, Ren S, Sun J (2016) Deep residual learning for image recognition. In: Proceedings of the IEEE Computer Society Conference on Computer Vision and Pattern Recognition. IEEE Computer Society, pp 770–778
9. Kondratyuk D, Yuan L, Li Y, Zhang L, Tan M, Brown M, Gong B (2021) MoViNets: Mobile Video Networks for Efficient Video Recognition. Proceedings of the IEEE Computer Society Conference on Computer Vision and Pattern Recognition 16015–16025
10. Gaikwad A, Williams RJ, de Winton H, Bevans BD, Smoqi Z, Rao P, Hooper PA (2022) Multi phenomena melt pool sensor data fusion for enhanced process monitoring of laser powder bed fusion additive manufacturing. *Mater Des* 221:110919
11. Hooper PA (2018) Melt pool temperature and cooling rates in laser powder bed fusion. *Addit Manuf* 22:548–559
12. Gaikwad A, Williams RJ, de Winton H, Bevans BD, Smoqi Z, Rao P, Hooper PA (2022) Multi phenomena melt pool sensor data fusion for enhanced process monitoring of laser powder bed fusion additive manufacturing. *Mater Des* 221:110919
13. Matsunawa A, Kim J-D, Seto N, Mizutani M, Katayama S (1998) Dynamics of keyhole and molten pool in laser welding. *J Laser Appl* 10:247–254
14. Cunningham R, Zhao C, Parab N, Kantzos C, Pauza J, Fezzaa K, Sun T, Rollett AD (2019) Keyhole threshold and morphology in laser melting revealed by ultrahigh-speed x-ray imaging. *Science* (1979) 363:849–852

REVIEWERS' COMMENTS

Reviewer #1 (Remarks to the Author):

The authors have made a substantial effort to address the concerns raised in the original reviews. However, there are still some sections of poor English that need to be revised. In addition to the comments below, the manuscript would benefit from being checked by a proofreader to improve readability.

pg5 ln 21: "In conducted experiments" change to "In the experiments" or remove completely

pg 7 ln 18: What are "Powder single beads"? Do you mean single track experiments?

pg8 ln 22: "Ti-6Al-4V are found to be not as good as that of other alloys as it has much different thermo-physical properties than the other alloys" needs revision. "not as good" is too vague for scientific writing and "has much different" doesn't make sense grammatically.

Reviewer #2 (Remarks to the Author):

The authors' rebuttal and revision have successfully addressed my review questions and concerns.

Response Letter

We highly appreciate the feedback and comments we received from the reviewers. We have answered the reviewers' questions and addressed their concerns. The manuscript has been revised accordingly. Our responses to the comments are provided below.

Reviewer #1:

The authors have made a substantial effort to address the concerns raised in the original reviews. However, there are still some sections of poor English that need to be revised. In addition to the comments below, the manuscript would benefit from being checked by a proofreader to improve readability.

pg5 In 21: "In conducted experiments" change to "In the experiments" or remove completely

pg 7 In 18: What are "Powder single beads"? Do you mean single-track experiments?

pg8 In 22: "Ti-6Al-4V are found to be not as good as that of other alloys as it has much different thermo-physical properties than the other alloys" needs revision. "not as good" is too vague for scientific writing and "has much different" doesn't make sense grammatically.

Response to Reviewer 1:

We appreciate the comments and feedback. We did a thorough proofreading and improved the readability of these sections.

1. *pg5 In 21: "In conducted experiments" change to "In the experiments" or remove completely*

We removed it.

2. *pg 7 In 18: What are "Powder single beads"? Do you mean single track experiments?*

That is correct. We changed it to "single tracks".

3. *pg8 In 22: "Ti-6Al-4V are found to be not as good as that of other alloys as it has much different thermo-physical properties than the other alloys" needs revision. "not as good" is too vague for scientific writing and "has much different" doesn't make sense grammatically.*

We have corrected the grammar of the sentence. We changed it to: "Results obtained from training on the Ti-6Al-4V data tend to be inferior to those obtained from training on the other alloy datasets, as the thermophysical properties of this metal alloy are significantly different from those of the other alloys"

Reviewer #2:

The authors' rebuttal and revision have successfully addressed my review questions and concerns.

Response to Reviewer 2:

We appreciate the comments and feedback.